# RETRIEVAL AUGMENTED IMPUTATION USING DATA LAKE TABLES

## ABSTRACT

Data imputation is an essential problem in many data science applications. Existing methods often struggle to impute missing values in scenarios where there is a lack of sufficient data redundancy. In this paper, leveraging large language models (LLMs) and data lakes, we propose a novel approach for retrieval-augmented imputation called **RAI**, utilizing fine-grained tuple-level retrieval instead of traditional coarse-grained table-based retrieval. **RAI** addresses the challenges of retrieving relevant tuples for missing value imputation from a data lake, where tuples have heterogeneous attributes, diverse values, and missing values. Rather than simply searching for similar tables, **RAI** employs a tuple encoder to learn meaningful representations for capturing tuple similarities and differences, enabling effective identification of candidate tuples. The retrieved results are further refined by a tuple reranker. We also introduce a new benchmark, mvBench, to advance further research. We conduct extensive experiments, demonstrating that RAI significantly outperforms state-of-the-art table-based retrieval-augmented imputation methods by **10.7%**.

## 1 INTRODUCTION

Data quality is crucial for effective data analysis, with missing values being a common issue (Abedjan et al., 2016). These arise from reasons including undefined values, collection errors, and errors in SQL joins while merging datasets. Excessive missing values can significantly degrade the reliability of downstream applications (Chai et al., 2023) and decision-making processes (Luo et al., 2020). Therefore, numerous efforts have been made to address the problem of missing values (Abiteboul et al., 1995; Jerez et al., 2010; Mahdavi & Abedjan, 2020). We can classify existing solutions into two categories: *leveraging data redundancy in the table itself* or *leveraging external knowledge*, as shown in Table 1.

**Leveraging Data Redundancy.** Most existing solutions fall into this category. The presence of repeating or similar data within the table enables these methods to extract patterns, dependencies, and relationships, facilitating missing value imputation. However, there is no one-size-fits-all solution for imputation, as different methods are tailored to different scenarios and types of missing values. For instance, this category is especially effective for continuous numerical data because the imputation heavily depends on the context within the table itself.

Nevertheless, these methods often fall short in scenarios that *lack sufficient data redundancy*. For example, small or sparse datasets, such as web tables, typically do not have enough similar data points to infer missing values. This challenge is further amplified in cases where each row contains highly diverse or unique information, making it difficult to apply patterns from one part of the table to another. Therefore, there is a growing need for methods that leverage external or domain-specific knowledge to compensate for the lack of data redundancy.

**Leveraging External Knowledge.** LLMs have emerged as a promising approach by utilizing their vast internal knowledge (Deng et al., 2022; Li et al., 2023). However, LLMs can suffer from issues like hallucinations and a lack of interpretability, making it difficult for users to trust and understand the imputed values. A potential solution to these challenges is Retrieval-Augmented Generation (RAG), which enhances LLMs by incorporating external data sources, offering more grounded and reliable imputation.

Table 1: A summary of imputation methods.

| Type | Category | Existing Work |
|---|---|---|
| **Leverage Data Redundancy in Table Itself** | Integrity Constraint | FD (functional dependency) (Abiteboul et al., 1995), CFD (Bohannon et al., 2007), RFD (Breve et al., 2022) |
| | Statistical Methods | Mean (Farhangfar et al., 2007), KNNI (Altman, 1992), CDI (Jerez et al., 2010) |
| | Machine Learning | MissFI (Stekhoven & Bühlmann, 2012), MICE (Royston & White, 2011), Baran (Mahdavi & Abedjan, 2020), HoloClean (Rekatsinas et al., 2017) |
| | Deep Learning | VAEI (McCoy et al., 2018), GAIN (Yoon et al., 2018), DataWig (Biessmann et al., 2019) |
| **Leverage External Knowledge** | LLMs | **Fine-Tuning** TURL (Deng et al., 2022), Table-GPT (Li et al., 2023) **In-Context Learning** GPT (Narayan et al., 2022) |
| | RAG for LLMs | **Table-based Retrieval** RATA (Glass et al., 2023) **Tuple-based Retrieval** Our Proposal: **RAI** |

Figure 1: Example of missing value imputation utilizing external knowledge.

**Challenges of RAG for Missing Value Imputation.** Despite the potential of RAG for data imputation, current methods face several limitations.

(C1) *The precision of retrieval and imputation*. Existing methods, *e.g.,* RATA (Glass et al., 2023), often operate at a coarse granularity by indexing and retrieving entire tables rather than fine-grained tuples. However, in practice, imputing missing values for a single tuple often requires information from only a few relevant tuples, which may be scattered across different tables in a data lake.

Figure 1 illustrates this, where filling in the missing "district", "party", and "birth date" values for the incomplete tuple requires gathering specific facts from multiple tables. Moreover, even when a relevant table is retrieved, *precisely locating the useful tuples* is still necessary. In the example, table T1 contains 400+ tuples, but only the matching "Bob Riley" tuple provides the information needed. Our experiments show that inputting excessive irrelevant information can actually degrade the imputation model's performance.

(C2) *Handling heterogeneous data*. Data lakes often contain heterogeneous data sources with varying schemas, missing values, and textual representations. For example, in Figure 1, T1, T2, and T3 have schema heterogeneity, with the attributes "Representative", "Governor", and "Incumbent" having the same semantic meaning. The tables also contain missing values, *e.g.,* "Rank" in T1. Furthermore, "R" in T1 and "Republican" in T3 illustrate variations in textual representations. This heterogeneity challenges retrieval mechanisms in aligning and comparing tuples for imputation.

**Contribution.** We introduce a novel approach for RAG-based missing value imputation at the tuple level within a data lake. Our main contributions can be summarized as follows:

- We employ contrastive learning and synthesized training data to learn tuple embeddings that capture similarities across heterogeneous data. This approach enables efficient similarity searches among tuples with diverse schemas and values, addressing the challenge of handling heterogeneous data (Challenge 1).

- We introduce **RAI**, a tuple-level retrieval-augmented framework that retrieves the top-$K$ relevant tuples from the data lake for a given incomplete tuple, reranks them to select a compact subset of top-$k$ tuples, and employs LLMs for accurate and context-aware imputation. By efficiently retrieving, reranking, and leveraging relevant tuples for imputation, **RAI** addresses the challenge of precision in retrieval and imputation (Challenge 2).

- We propose a large-scale benchmark, `mvBench`, with **15,143** incomplete tuples, **4.23** million tuples within the data lake. For each incomplete tuple, the relevant tuples are labeled manually, enabling a fine-grained evaluation of RAG-based data imputation.

- We conduct extensive experiments on `mvBench` and compare **RAI** with thirteen baseline methods to showcase its effectiveness in imputing missing values in small tables. Our results indicate that **RAI** significantly outperforms state-of-the-art table-based retrieval-augmented imputation methods by **10.7%**. Our code and data are open-source[1].

---

[1] https://anonymous.4open.science/r/Retrieval_Augmented_Imputation-D376

## 2 SOLUTION OVERVIEW

### 2.1 PROBLEM STATEMENT

A data lake $L = \{D_1, D_2, \ldots, D_k\}$ is a collection of $k$ tables, each of which may have a distinct schema. A relational table $D$ consists of a schema, which is a set of attributes $R(D) = \{A_1, A_2, \ldots, A_n\}$, defining the columns of the table, a set of tuples $\{t_1, t_2, \ldots, t_m\}$, and a textual caption. An incomplete tuple $t$ is a tuple that contains one or more missing values. Given an incomplete tuple $t$ and a data lake $L$, the problem of *missing value imputation using data lakes* involves repairing $t$ by retrieving relevant tuples from $L$. The goal is to ensure that the repaired tuple $t$ closely resembles its ground truth counterpart $t_g$.

### 2.2 THE RAI FRAMEWORK

We adopt a Retrieve-Rerank-Reason RAG framework for data imputation, namely **RAI**. An illustrative example is shown in Figure 2, which demonstrates how to fill in the missing team for player Adrian Aucoin in the 2012-13 NHL season. The process is broken down as follows: Given an incomplete tuple $t$, the **Retriever** initially retrieves the top-$K$ tuples from the data lake that are most relevant to $t$. In the example, the tuple ranked #1 lacks the team attribute, while the tuple ranked #2 lists all the teams Aucoin has played for, but without specifying the time periods. Only the tuple ranked #20, *i.e.,* the relevant tuple, records Aucoin's team change in the NHL in 2012. Subsequently, the **Reranker** reranks these top-$K$ tuples through a fine-grained comparison between $t$ and each retrieved tuple. The most relevant tuple is then elevated to rank #2 in this example. Finally, the top-$k$ ($k = 5$ in the example) retrieved tuples are then provided to the **Reasoner** for imputing the missing value rationally.

To effectively implement **RAI** for data imputation using data lakes, we face several key challenges:

(1) Retrieving relevant tuples from heterogeneous data for impution: Existing tuple embedding methods (Tang et al., 2021) fail to capture complex connections between an incomplete tuple and its relevant tuples. Entity matching techniques (Wang et al., 2023; Li et al., 2020) are also inadequate, as they are domain-specific, with each dataset consisting of two tables within the same domain. Thus they cannot handle diverse and heterogeneous data in data lakes. Moreover, identifying a relevant tuple for imputation differs from finding a matched tuple in entity matching, as in Figure 1, the relevant tuple in T3 does not describe the same entity as the incomplete tuple.

(2) Precisely identifying relevant tuples: Previous retrieval-augmented imputation methods (Glass et al., 2023) lack precision in identifying relevant tuples, hindering the imputation process and burdening the reasoner.

(3) Enhancing reasoning with domain knowledge: Data imputation requires the reasoner to possess domain knowledge and reasoning capabilities beyond simply aligning attributes and extracting values (Zhang & Balog, 2019; Glass et al., 2023). For instance, in Figure 1, understanding that the "party" is about "Siegelman" but not "Reiley" in T3 is necessary for accurate imputation. Similarly, in Figure 2, the reasoner must recognize that the "teams" in the second tuple describes all the teams the player has been in and cannot be used as evidence for imputation of the team the player was on during the 2012-2013 season.

In the following sections, we will discuss how each module in **RAI** is designed to address these challenges, enabling effective data imputation using data lakes.

## 3 RETRIEVAL AUGMENTED IMPUTATION

### 3.1 RETRIEVER

One promising approach for handling heterogeneous data is to transform all tuples into embedding vectors through a tuple encoder, **enc**($\cdot$). To achieve this, our basic idea is to train a tuple encoder, **enc**($\cdot$), which ensures that embeddings of a tuple $t$ with missing values and its relevant tuple $s$ from a data lake are similar, *i.e.,* **enc**($t$) $\approx$ **enc**($s$). Conversely, embeddings of tuples irrelevant to $t$ will be significantly distinct.

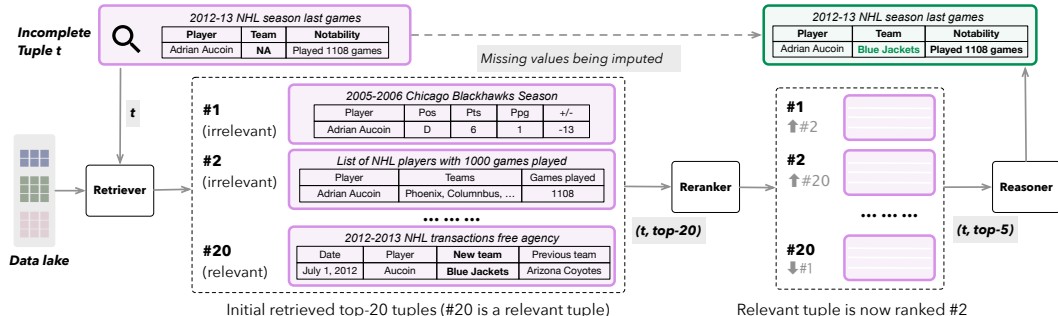

Figure 2: A Running Example of the Retriever/Reranker/Reasoner Framework. (Note: for simplicity, we omit the caption of the table.)

To better capture pairwise tuple relationships, we employ contrastive learning within a Siamese network architecture. Despite the potential of it, there are still two main challenges. First, the heterogeneous nature of tuples in data lakes poses several obstacles in learning effective representations: attribute heterogeneity due to varying schemas or formats, textual representation variance caused by synonyms and abbreviations, and the presence of missing values complicate the learning of effective representations and the comparison process. Furthermore, obtaining sufficient and diverse training data is another significant challenge in learning tuple representations. Next, we will discuss how to learn a tuple encoder with contrastive learning and how to synthesize diversified training data to solve the above two challenges.

### 3.1.1 CONTRASTIVE LEARNING FOR TUPLE ENCODING.

We utilize contrastive learning within a Siamese network (Chopra et al., 2005), characterized by its dual-encoder structure and shared weights (Reimers & Gurevych, 2019). Each training sample is a pair consisting of an "anchor" tuple and either a "positive" or a "negative" tuple. A batch $\mathcal{B}$ consists of training examples for $N$ anchors, $x_i$ ($i = [1, N]$). For anchor $x_i$, we construct one positive pair $(x_i, y_i^+)$ and $M$ negative pairs $(x_i, y_{i,j}^-)$ ($j = [1, M]$). We denote $Y$ as $\sum_{i=1}^{N}(y_i^+ + \sum_{j=1}^{M} y_{i,j}^-)$. In the training process, we employ the in-batch negative strategy, since previous work (Karpukhin et al., 2020) has demonstrated that increasing the number of negatives can improve retrieval performance. We optimize the following contrastive loss function to maximize the similarity between positive pairs while minimizing the similarity between negative pairs.

$$\mathcal{L} = -\frac{1}{N} \sum_{i=1}^{N} \log \frac{\exp(sim(x_i, y_i^+))}{\sum_{y_k \in Y} \exp(sim(x_i, y_k))}$$

where $sim(x, y) = enc(x) \cdot enc(y)$, which calculates the similarity between the embeddings of the anchor tuple $x$ and either the positive or negative $y$. We use dot product as the similarity function.

### 3.1.2 SYNTHESIZING TRAINING DATA.

Effectively training a tuple encoder with contrastive learning requires extensive training data. However, the lack of suitable datasets and the high cost of manual labeling (e.g. MS MARCO (Nguyen et al., 2016)) necessitate the automatic synthesis of training data. To fill in this gap, we propose a novel approach that employs tuple augmentation operators and a three-step process for synthesizing training data. This approach also addresses key challenges tuple encoder faces, such as attribute heterogeneity, diverse values, which we will demonstrate in the following.

**Tuple Augmentation Operators.** We first introduce tuple augmentation operators that transform a tuple into its "equivalent" form. These systematically designed operators will be used to construct training data that simulates the diverse and heterogeneous nature of data lakes.

Our data augmentation operators in Table 2 are designed to augment a tuple from three aspects: caption, attribute, or value. These operators can help capture various forms of relevant tuples originating from diverse sources, enhancing flexibility and robustness in tuple retrieval and comparison. For instance, value augmentation operators generate equivalent tuples with synonymous or missing values, mimicking the variance in textual representations and the presence of missing values. For

specific examples of each category, please refer to the "Example" column in Table 2. Note that our augmentation strategy deliberately avoids insertion operations to prevent excessive noise introduction and uses the "replace_val" operator cautiously, only in instances where the cell contains synonyms, which are obtained from values associated with the same entity across the dataset.

Then we explain the process of constructing anchor tuples and generating their corresponding positive and negative pairs using these augmentation operators. In Appendix D, we provide examples of the training data to present this process more clearly.

Table 2: Tuple augmentation operators.

| Consider a sample tuple with a caption and attribute/value pairs: Caption: "Harrisburg, Pennsylvania, Sports" (**club:** Harrisburg . . ., **league:** USL Soccer, **venue:** Skyline . . .) | | |
|---|---|---|
| **Type** | **Operator** | **Example** |
| Caption | delete_cap | Harrisburg, NA, Sports |
| | replace_cap | Harrisburg, Pennsylvania, Athletic |
| | shuffle_cap | Pennsylvania, Harrisburg, Sports |
| Attribute | shuffle_att | new attribute order: (**league, venue, club**) |
| | delete_att | new attribute set: (**club, venue**) |
| Value | replace_val | Harrisburg, United Soccer League, . . . |
| | empty_val | Harrisburg, NA, Skyline |

**Synthesizing Anchor Tuples.** We utilize the WikiTables-TURL dataset (Deng et al., 2022) to construct two types of anchor tuples: (1) complete tuples without missing entries, enabling the model to learn the full semantic information of tuples; and (2) tuples with 30% of significant cells (*e.g.,* country names like "USA") masked with [MASK] symbol, helping the model learn the intent of imputation. We synthesize anchor tuples in a 70/30 ratio of complete to masked tuples, which our experiments show yields superior results.

**Synthesizing Positive Tuples.** To create positive samples for an anchor tuple $x_i$, we employ two strategies: (1) augmenting $x_i$ using designed tuple augmentation operations to generate positive tuples while maximizing the diversity; and (2) identifying tuples from other tables that share the same subject entity with $x_i$ through entity linking, then augmenting those tuples. If the anchor tuple contains masked cells, we ensure that the positives contain the content of a masked cell, reinforcing the model's ability to identify tuples that can successfully impute missing information. The latter strategy addresses attribute heterogeneity and textual variance, simulating the common scenario where relevant data sources vary widely. These strategies inject heterogeneity between positive and anchor tuples, enabling the model to handle heterogeneous data for data imputation effectively.

**Synthesizing Negative Tuples.** We generate negative samples for each anchor tuple in two categories: easy and hard negatives. Easy negatives are randomly selected tuples from other tables, due to the in-batch negative strategies, we do not need to construct them deliberately. Hard negatives, on the other hand, are selected from the same table as the anchor tuple but represent different entities. This distinction is essential as it forcing the encoder to learn more discriminative and meaningful representations, enabling it to distinguish between similar but distinct tuples (Luo et al., 2023). All negative samples are augmented for increasing diversity. By synthesizing both easy and hard negatives, we create an informative and diversified training dataset. This dataset enables the model to effectively recognize negative (*i.e.,* irrelevant) tuples to a given incomplete tuple.

**Training Data Summary.** In summary, for our retriever, the training set comprises **282,862** anchor tuples from **41,260** tables, and the development set includes **9,460** anchor tuples from **775** tables. Each anchor tuple is paired with 1 positive and 7 negative tuples. Notably, our encoder, trained on this dataset, has shown strong generalization capabilities, performing well on larger downstream datasets that contain an even greater volume of table data.

**Indexing.** We employ the vector database Meta Faiss (Johnson et al., 2019) to encode all tuples into 768-dimensional vectors using Faiss's flat indexing system, which compresses the vectors into fixed-size codes that are stored in an array.

## 3.2 RERANKER: A DESIGN SPACE EXPLORATION

After obtaining the top-$K$ tuples from the retriever, we find that relevant tuple is often retrieved only when $K$ is sufficiently large (*e.g.,* $K = 100$), increasing the complexity of the subsequent reasoning. To mitigate this issue, we introduce a fine-grained reranker component. However, despite the proven effectiveness of reranking in text retrieval, the optimal reranker for retrieval-augmented imputation remains unclear due to the lack of comprehensive exploration. To address this gap, we conduct an extensive investigation into the design space of reranking methods for calculating the relevance of a retrieved tuple $s$ from the retriever with respect to an incomplete tuple $t$. We

Table 3: Statistics of mvBench (Tab. : Tables; Tup. : Tuples; Attrs: Attributes).

| Datasets | Incomplete Tuples | | Data Lake | | #-Relevant Tup. | Part of | #-Training |
|          | #-Tab. | #-Tup | #-Tab. | #-Tup | / #-Tup. | Missing Attrs | Tup. |
|---|---|---|---|---|---|---|---|
| WikiTuples (WT) | 807 | 10,003 | 207,912 | 2,674,164 | 4.38 | Party, Director, ... | 100 |
| Show Movie (SM) | 1 | 30 | 2 | 19,586 | 1 | Age Rating | 6 |
| Cricket Players (CP) | 1 | 213 | 2 | 94,164 | 1.38 | Batting Style, ... | 20 |
| Education (ED) | 2 | 654 | 17 | 11,132 | 4 | Address, Phone, ... | 30 |
| Business (BU) | 1 | 4,243 | 3 | 1,436,951 | 2.62 | City, ZipCode, .. | 100 |

categorize existing methods into two main categories: fine-tuning methods and prompting methods, as outlined in previous work (Zhu et al., 2023).

**Fine-tuned methods** involve fine-tuning a language model to enhance reranking capabilities, which can be further subdivided into scoring-based rerankings (Nogueira & Cho, 2019; Gao et al., 2021) that compute a numerical relevance score for each (incomplete tuple $t$, retrieved tuple $s$) pair, and generative relevance reranking (Nogueira et al., 2020) that outputs a "true" or "false" token, indicating the relevance between $t$ and $s$.

**Prompting models**, on the other hand, send prompts to LLMs without fine-tuning. These methods can be classified as pointwise (Sachan et al., 2022), which evaluates the relevance of a (incomplete tuple $t$, retrieved tuple $s$) pair individually; listwise (Sun et al., 2023; Ma et al., 2023), which assesses and ranks an entire list of retrieved tuples ($s_1, s_2 \ldots, s_k$) collectively; and pairwise (Qin et al., 2023), which compares a set of pairs (incomplete tuple $t$, retrieved tuple $s_i$) to ascertain which $s_i$ is more relevant to $t$.

Previous research (Sun et al., 2023; Ma et al., 2023) has indicated that the pointwise method performs poorly compared to listwise and pairwise methods. Therefore, we focus on the remaining methods, choosing the most widely recognized model framework for each. To our knowledge, this is the first study on using prompting methods for reranking in tuple retrieval. Our empirical results show that fine-tuned generative relevance reranking model performs best when trained on a dataset of no more than 100 incomplete tuples. Next, we will discuss the best-performing model, while the other models and experimental results are discussed in Section 4.

**Our Default Reranker.** Following the generative relevance reranking approach (Nogueira et al., 2020), we construct $K$ pairs of an incomplete tuple $t_i$ and its top-$K$ retrieved tuples from retriever. Each pair is serialized and concatenated into a sequence, which is then input into a seq2seq model, typically a T5-base model (Raffel et al., 2020a). The model outputs a single token, and a softmax function is applied to the logits associated with the "true" and "false" tokens to generate a relevance score, thus obtaining the likelihood of a retrieved tuple being relevant to the anchor. We then order the top-$K$ retrieved tuples based on these scores to rank their relevance and get top-$k$ reranked tuples.

### 3.3 REASONER

After obtaining the top-$k$ reranked results, the process proceeds to the crucial data imputation stage. We leverage LLMs to perform the final reasoning step, using structured prompts to guide them in generating the desired outputs. The template we used and example of the reasoning process are provided in the Appendix C. Note that when an incomplete tuple contains multiple missing values, imputation can be performed as long as relevant tuples for filling those missing values are retrieved, as demonstrated in the example.

## 4 EXPERIMENT

Our experiments aim to answer three key questions: (1) How does **RAI** compare to other methods in terms of end-to-end imputation performance? (Section 4.3) (2) How effectiveness of our retriever (Section 4.4)? (3) Which reranker is most suitable for data imputation to explore the factors influencing **RAI** (Section 4.5)?

### 4.1 DATASET

Although existing datasets (Mei et al., 2021; Mahdavi & Abedjan, 2020; Glass et al., 2023) for data imputation provide missing data and their corresponding ground truth, they mostly lack: (1) large

data lakes containing massive tables to assist in filling missing values, and (2) labeled relevant tuples or tables for imputation. To address these limitations, we introduce **mvBench**, a large-scale benchmark with 15,143 incomplete tuples and 4.23 million tuples from the data lake for missing value imputation. We also provide relevant tuples annotated by human experts for each incomplete tuple to enable a fine-grained evaluation of the retrieval module. Our benchmark focuses on challenging scenarios that require external sources for imputation, which is not well addressed by existing work that typically rely on the inherent data redundancy within the table. By including these scenarios, we demonstrate the effectiveness and adaptation of our tuple-level RAG approach. The detailed construction of datasets is presented in Appendix F.

**mvBench** comprises five datasets collected from real-world scenarios, varying in scales, domains, and sources. Table 3 presents detailed statistics of all datasets within **mvBench**. For each dataset, a subset of incomplete tuples is randomly sampled to form the training set for the reranker (see "#-Training Tup." column of Table 3), while the remaining serves as the test set for evaluation. Our retriever is trained only on the synthesized data and can be directly applied to the aforementioned datasets without additional training.

## 4.2 EXPERIMENTAL SETTINGS

We introduce baselines and evaluation metrics for the reasoner, retriever, and reranker components of **RAI**. Detailed hyperparameters and environment settings are provided in Appendix A.

**Baselines for Reasoner (Data Imputation).** We compare **RAI** with existing imputation solutions that leverage external knowledge beyond the table itself, including LLM with in-context learning, LLM with BM25 retriever, LM with fine-tuning TURL (Deng et al., 2022), and table-based retrieval-augmented imputation RATA (Glass et al., 2023). As mentioned in Section 1, we focus on the scenario where data redundancy within the table is lack, and thus do not include comparisons with methods that primarily rely on it. For retrieval-based methods, we send the top-5 retrieved results along with incomplete tuples to LLMs for final imputation. We run the first two baselines on our benchmark and compare them with **RAI**, while the comparisons with TURL and RATA are discussed separately due to differences in their settings.

**Baselines for Retriever.** We compare our retriever against five baselines: (1) BM25 (Robertson et al., 2009), (2) Contriever (Izacard et al., 2021), and (3) DPR-scale (Lin et al., 2023), all of which have zero-shot capabilities and excel in few-shot and zero-shot passage retrieval; (4) a BERT-based tuple encoder with masked language modeling (MLM) where 30% of tuple cells are randomly masked, to compare with previous works on tuple representation that only accept individual tuples as input and adopt pre-training tasks centered around language modeling (Tang et al., 2021); and (5) Sudowoodo (Wang et al., 2023), a state-of-the-art entity matching method. For each dataset, following its original settings, we randomly select 10,000 tuples from incomplete tuples and the data lake as the labled data, then pretrain the model for 3 epochs and finetune it for 40 epochs. After that, Sudowoodo can encode tuples into vectors, enabling retrieval using FAISS. Although Sudowoodo can impute missing values, it relies on redundant information within the table itself, which differs from our scenario.

**Baselines for Reranker.** Our default reranker is a fine-tuned generative relevance reranking model, and we compare it with several baselines, including fine-tuned scoring-based methods: RoBERTa-LCE (Gao et al., 2021) and monoBERT (Nogueira & Cho, 2019), as well as prompting methods using GPT-3.5 with listwise (Sun et al., 2023; Ma et al., 2023) and pairwise reranking (Qin et al., 2023). In Appendix B, we provide the implementation details for those reranking methods.

**Evaluation Metrics.** We evaluate the end-to-end performance of RAI for data imputation using Exact Match (EM) Accuracy(Izacard & Grave, 2020), which considers a generated value correct if it matches any acceptable answer after normalization. The performance of the retriever and reranker are assessed using recall@K and success@K respectively.

## 4.3 EVALUATION FOR DATA IMPUTATION

**Main Results.** Table 4 shows that **RAI** significantly improves imputation accuracy over using LLMs alone. LLMs often produce sub-optimal results due to limited accuracy and uncertainty in their

stored knowledge, even within the Wikipedia domain. In specific domains like Business and Education, both GPT-3.5 and GPT-4 struggle, with accuracies as low as 0.017 and 0.128 for GPT-3.5, and 0.598 and 0.114 for GPT-4, respectively. In contrast, **RAI** demonstrates markedly superior performance by integrating advanced reasoning capabilities of LLMs with rich knowledge from retrieved tuples. Excluding outlier results from the Business and Education datasets, the average improvements for **RAI** over GPT-3.5 and GPT-4 are 28.00% and 13.84%, respectively.

The results also highlight the importance of an effective retriever. BM25 performs worse than **RAI** on most datasets, except for Business, where high lexical overlap favors BM25. (You can see BM25's excellent retrieving perfor-

Table 4: Experimental results of data imputation.

| Reasoner | Retriever | WT | SM | ED | CP | BU |
|---|---|---|---|---|---|---|
| | w/o | 0.715 | 0.875 | 0.017 | 0.896 | 0.128 |
| GPT-3.5 | w/ tuples(BM25) | 0.577 | 0.792 | 0.894 | 0.889 | **0.983** |
| | w/ tuples(RAI) | **0.866** | **0.875** | **0.976** | **0.964** | 0.98 |
| | w/o | 0.752 | 0.75 | 0.598 | 0.863 | 0.114 |
| GPT-4 | w/ tuples(BM25) | 0.8 | 0.875 | 0.925 | 0.909 | **0.998** |
| | w/ tuples(RAI) | **0.902** | **0.917** | **0.979** | **0.972** | **0.998** |

mance on the Business dataset in Table 5.) Using a sub-optimal retriever like BM25 can adversely impact the data imputation accuracy of a less advanced model like GPT-3.5, as inaccurately retrieved tuples can misguide LLMs with weaker inferencing capabilities. GPT-4 generally outperforms GPT-3.5, except on Cricket Players and Business datasets without retrieved tuples, where both models lack domain-specific knowledge. In this case, GPT-3.5 guesses an answer, while GPT-4 often provides no answer. However, the performance gap between GPT-3.5 and GPT-4 diminishes when retrieval-augmented imputation is used, indicating the advantage of our framework.

**Ablation Study.** We conduct two additional experiments to further explore the factors influencing the performance of our retrieval-augmented imputation framework. These experiments are detailed in Appendix E.1 and E.2.

The first experiment investigates how the number of retrieved tuples fed to the reasoner affects imputation accuracy. Surprisingly, sending more retrieved tuples to the reasoner does not guarantee better imputation accuracy due to the added complexity it introduces to the reasoning process, and in some cases, the performance even declines. This underscores the importance of an efficient retrieval module capable of achieving a high success rate with the smallest possible number of retrieved tuples, supporting our decision to select only the top-5 retrieved tuples and highlighting the importance of employing a reranker to optimize outcomes.

The second experiment investigates the impact of the number of example tuples (complete tuples from the same table as the incomplete tuple) sent to the reasoner on imputation performance. We hypothesize that providing more complete example tuples would enable LLMs to impute missing values more accurately by guiding the model towards the domain and format of the missing value. However, the results show no clear correlation between the number of example tuples and imputation accuracy. This might be attributed to the inherent lack of redundant data in the tables and LLMs' limited accurate knowledge for filling in missing values.

**Discussion on Other Data Imputation Methods.** We compare **RAI** to two types of data imputation methods: TURL (fine-tuned LM) and RATA (table-based RAG).

(1) **RAI** v.s. TURL (fine-tuned LM): We compare **RAI** against TURL (Deng et al., 2022) on the WikiTuples dataset, which is constructed from TURL's test set. It is important to note that TURL's performance on this dataset can be considered an upper bound, as all cells to be filled appear in TURL's training set at least three times and relations between different entities are learned. Although **RAI** achieves a slightly lower imputation accuracy compared to TURL (0.902 vs. 0.967), it is remarkable considering that **RAI**'s training data size is only $1/14$ of TURL's. Moreover, TURL requires that the ground truth of the cell to be filled must be linked to an entity in its pre-constructed entity vocabulary, and it can only output entity id from this vocabulary. In contrast, **RAI** is designed to complement LLMs and can be easily adapted to different datasets without constraints. This flexibility and adaptability make **RAI** a more robust solution for data imputation tasks across various domains and datasets.

(2) **RAI** (tuple-based RAG) v.s. RATA (table-based RAG): While RATA focuses on table retrieval, our dataset requires more fine-grained retrieval at the tuple level. Additionally, RATA's definition of the relevant table for an incomplete table is simply a table that contains the ground truth of the

Table 5: Performance of retriever (recall rate: R; success rate: S).

| Retriever | WT | | SM | | ED | | CP | | BU | |
|---|---|---|---|---|---|---|---|---|---|---|
| | R@100 | S@5 | R@100 | S@5 | R@100 | S@5 | R@100 | S@5 | R@100 | S@5 |
| BM25 | 0.327 | 0.286 | 0.792 | 0.5 | 0.743 | 0.901 | 0.902 | 0.739 | 1.0 | **1.0** |
| Contriever | 0.484 | 0.455 | 0.042 | 0.0 | 0.758 | 0.825 | 0.074 | 0.043 | 0.006 | 0.001 |
| DPR-scale | 0.497 | 0.253 | 0.458 | 0.167 | 0.143 | 0.016 | 0.048 | 0.005 | 0.035 | 0.004 |
| BERT with MLM task | 0.262 | 0.203 | 0.0 | 0.0 | 0.0 | 0.0 | 0.0 | 0.0 | 0.0 | 0.0 |
| Sudowoodo | 0.727 | 0.506 | 0.417 | 0.125 | 0.974 | 0.617 | 0.987 | 0.926 | 1.0 | 0.972 |
| **Retriever** (ours) | **0.945** | **0.813** | **1.0** | **0.875** | **0.992** | **0.923** | **1.0** | **0.989** | **1.0** | 0.999 |

missing value. Thus, it's hard to adapt RATA to our `mvBench`. To provide a comparison, we evaluate **RAI** on RATA's EntiTables dataset, following the settings of RATA. **RAI** outperforms RATA in both retrieval (MRR@10 of 0.552, 47.28% improvement) and imputation accuracy (0.415, 10.7% improvement), attributed to **RAI**'s more refined tuple-based retrieval system. Despite requiring more storage space than RATA (30G *v.s.* 14G), **RAI** justifies its larger footprint by providing markedly better results. Note that the improvement in imputation accuracy is not as significant as the retrieval performance because RATA's definition of relevant tuples is simplistic, labeling tables containing the missing value as relevant without considering reasonable inference. For example, in Figure 2, a table with #2 tuple would also be labeled as relevant. However, LLMs can filter out many "relevant tuples" that do not lead to the correct answer.

## 4.4 EVALUATION FOR RETRIEVER

**Main Results.** Our retriever, leveraging contrastive learning, outperforms baselines in both recall and success rate across all datasets, demonstrating its superior effectiveness and generalization. Table 5 shows that our retriever achieves the highest recall and success rates on all datasets except Business. Notably, our retriever is pre-trained on 40k Wikipedia tables and directly applied to the five datasets without additional training, showcasing its robustness and generalization capability. The results also indicate that: (1) Retrievers designed for passage retrieval tasks cannot be directly applied to our task. (2) Contrastive learning is crucial for an effective tuple encoder for retrieval. **RAI**'s retriever and BERT with MLM task share the same base model and training corpus, but the latter performs worst across all datasets, indicating its unsuitability for our scenario. (3) Our task differs from entity matching, as tuples describing the same entity are not necessarily relevant tuples. This is evident from the fact that our retriever significantly outperforms Sudowoodo, which uses labeled data, especially on the WikiTuples and Show Movie datasets. In contrast, Sudowoodo performs well on Business and Cricket Players datasets since they contains many cases where the relevant tuple and the incomplete tuple describe the same entity (one of the cases in our scenario).

**Ablation Study.** To investigate the effectiveness of our synthesizing training data and identify key factors in its construction, we conduct comparative experiments focusing on anchor tuples, positives, and negatives. The experiments reveal that combining complete anchor tuples with those having missing values enhances retriever performance, as including missing values aligns with the intent of data imputation, while complete anchors help the retriever learn tuple structure and overall semantics. Additionally, including diverse positive samples with various heterogeneous attributes from other tables significantly improves the retriever's performance, aligning with real-world scenarios. Moreover, intuitively, treating anchors with the masked missing cell deleted as negatives should help the model better understand that, if a tuple does not contain the content corresponding to the masked missing value, it cannot serve as a relevant tuple, even if it is very similar to the incomplete tuple. Surprisingly, adding these hard negatives leads to a significant decrease in retrieval results, possibly due to the difficulty in accurately capturing and distinguishing cell-level semantics when encoding tuples. In summary, the data synthesizing method for our retriever proves to be very effective, with the combination of anchor tuples, positives, and negatives being crucial. The details are provided in Appendix E.3.

## 4.5 EVALUATION FOR RERANKER

We compare fine-tuned and prompting methods for reranking. Fine-tuned methods use the complete test set, while prompt-based rerankers are evaluated on a sampled subset due to cost limitations. Results are presented in Table 6 (a) for the complete dataset and Table 6 (b) for the sampled subset.

Table 6: Performance of reranker (success rate: S).

| Reranker | WT | | SM | | ED | | CP | | BU | |
|---|---|---|---|---|---|---|---|---|---|---|
| | S@1 | S@5 | S@1 | S@5 | S@1 | S@5 | S@1 | S@5 | S@1 | S@5 |
| (a) results on full test set | | | | | | | | | | |
| Initial Retrieval | 0.534 | 0.813 | **0.792** | **0.875** | 0.577 | 0.923 | 0.83 | 0.989 | 0.033 | **1.0** |
| monoBERT | 0.553 | 0.799 | 0.667 | **0.875** | 0.973 | 0.984 | 0.622 | 0.931 | 0.992 | 0.991 |
| RoBERTa-LCE | 0.654 | 0.904 | 0.083 | 0.417 | 0.97 | 0.984 | 0.902 | 0.974 | 0.998 | **1.0** |
| **Reranker** (ours) | **0.754** | **0.926** | 0.708 | **0.875** | **0.976** | **0.986** | **0.941** | **1.0** | **0.999** | **1.0** |
| (b) results on partial test set | | | | | | | | | | |
| Initial Retrieval | 0.465 | 0.74 | 0.792 | **0.875** | 0.655 | 0.975 | 0.852 | 0.989 | 0.095 | 0.99 |
| GPT-3.5 w/ Pairwise | 0.475 | 0.76 | **0.833** | **0.917** | 0.26 | 0.96 | **0.932** | 0.989 | 0.695 | 0.99 |
| GPT-3.5 w/ Listwise | 0.3 | 0.68 | 0.708 | 0.833 | 0.255 | 0.825 | 0.75 | 0.841 | 0.37 | 0.945 |
| **Reranker** (ours) | **0.71** | **0.905** | 0.708 | 0.875 | **0.995** | **0.995** | 0.932 | **1.0** | **0.995** | **1.0** |

**RAI**'s reranker (generative relevance reranking) achieves the highest success@5 compared to other fine-tuned rerankers across various datasets. Excluding Show Movie and Business datasets, **RAI**'s reranker improves success@1 by 41.2% and success@5 by 7.2% compared to our retriever's initial results. The reranker's failure on the Show Movie dataset is due to insufficient training data (only 6 incomplete tuples). For the Business dataset, while the initial success@1 is only 0.033, the success@5 reaches 1.0. This is mainly because the dataset contains many tuples describing the same company without missing information, and the retriever struggles to differentiate cell-level semantics while the reranker can distinguish this effectively, significantly improving success@1. Our reranker also outperforms existing prompting methods across datasets, except Show Movie. GPT-3.5 with listwise reranking is the least effective, with success@5 lower than **RAI**'s initial retrieval results, due to its limited understanding of tabular data and the challenge of reranking a large list of tuples directly. GPT-3.5 with pairwise reranking shows improvements, as choosing between two options is simpler than sorting an entire list. However, it still performs poorly compared to our reranker, since incorrectly ranking a relevant tuple lower in a pairwise comparison leaves little chance for its position to improve in subsequent sorting.

## 5 RELATED WORK

Data imputation without sufficient data redundancy has gained attention with the advancement of LLMs. While some studies have applied LLMs directly to data imputation through in-context learning (Narayan et al., 2022) or training models to understand tabular structure and knowledge (Li et al., 2023; Zhang et al., 2023), ensuring high accuracy and reliability remains a challenge. Retrieval-Augmented Generation (RAG), introduced by (Lewis et al., 2020), involves retrieving relevant documents from external sources to generate answers. RAG has been adopted for table-related tasks, such as TableQA (Herzig et al., 2021) and table augmentation (Glass et al., 2023), but the utilization of RAG in data imputation remains relatively unexplored.

Previous imputation works incorporating retrieval ideas have limitations in addressing the challenges of imputing tables with limited redundancy. Some approaches retrieve information from the same table (Li et al., 2015) or use simple matching (Zhang & Balog, 2019), still relying on the table's inherent redundancy and failing to handle heterogeneous data. Others utilize external sources like master data (Fan et al., 2012; Interlandi & Tang, 2015) or knowledge bases (Hao et al., 2017; Chu et al., 2015), but require expert involvement, making them unsuitable for large-scale data lakes. RATA (Glass et al., 2023) employs a table-level retrieval framework for data imputation, but its coarse-grained retrieval is insufficient for accurate imputation in tables lacking data redundancy.

## 6 CONCLUSION

We introduce a Retrieval-Augmented Imputation Framework called **RAI**, specifically tailored for addressing missing value imputation in data lakes. **RAI** integrates a pre-trained retriever capable of identifying relevant tuples, a fine-tuned reranker to ascertain fine-grained relevance, and a reasoner that applies in-context learning for the reliable imputation process. Our experiments demonstrate that **RAI** features an exceptionally effective retrieval module, surpassing various established baselines and markedly improving upon methods that depend solely on LLMs.

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

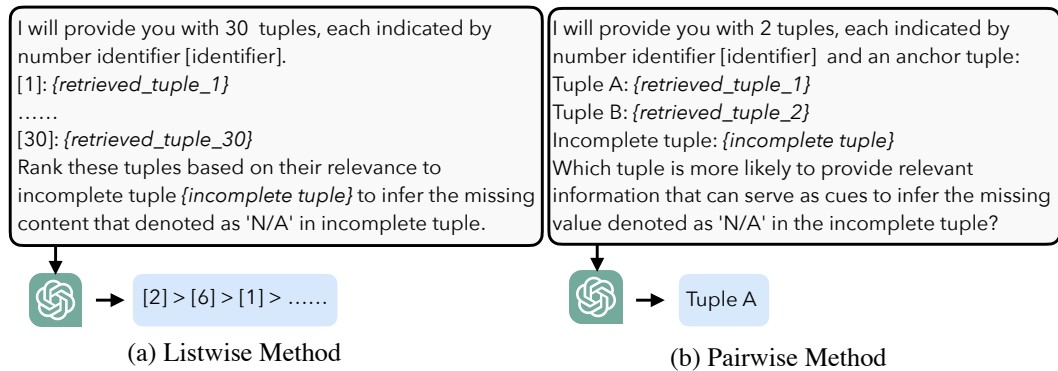

(a) Listwise Method  (b) Pairwise Method

Figure 3: Template for prompt-based reranking

## A EXPERIMENTAL SETUP

For retriever, we employ BERT-base-uncased (110M) [2] to initialize the model parameters. We set the batch size to 16 and the total training epochs to 2 and take AdamW (Loshchilov & Hutter, 2017) as the optimizer. The retriever training took approximately 7 hours on 4 RTX 4090 GPUs. We save the model every 10,000 step and select the one with the smallest loss on the development set. For reranker, we adopt T5-base model (Raffel et al., 2020b) and initialize it with monoT5 [3](220M) trained on the MS MARCO passage dataset (Nguyen et al., 2016). For reasoner, we use gpt-35-turbo-1106 (GPT-3.5) and gpt-4-1106-preview (GPT-4) with the temperature of 0.3. All experiments are run on an Ubuntu 22.04 server with 8 RTX 4090 GPUs.

## B IMPLEMENTATION DETAILS OF RERANKING METHODS

### B.1 FINE-TUNED RERANKING

For the training data of the fine-tuned reranker, we use a randomly sampled subset of incomplete tuples from each dataset (see the "#-Training Tup." column of Table 3). We prepare training data consisting of (Incomplete, Positive/Negative) pairs. For each incomplete tuple, the relevant tuples that have already been labeled are used as positive samples. Negatives are randomly selected from the top-20 results retrieved by our retriever with additional filtering to ensure they are not relevant to the incomplete tuple.

### B.2 PROMPTING-BASED RERANKING

**GPT-3.5 with Listwise Reranking.** The listwise method utilizes a strategy where the LLM is prompted with a query and a document list and is asked to output the identifiers of the documents in a reranked order based on their relevance to the query (Sun et al., 2023; Ma et al., 2023). As depicted in Figure 3 (a), in our scenario, this method is implemented by feeding LLMs with a set of retrieved tuples and an incomplete tuple, each retrieved tuple is paired with an identifier. The model then generates a list of reranked identifiers according to the relevance to the query tuple. Due to input size limitations of the LLM, a sliding window strategy is employed, with a window size of 30 and a step size of 14, to manage larger lists. This method is supported by the work in (Sun et al., 2023; Ma et al., 2023).

**GPT-3.5 with Pairwise Reranking.** In addition to listwise reranking, Qin et al. (Qin et al., 2023) introduces a pairwise reranking method utilizing LLMs as rerankers. Similar to the bubble sort algorithm, this technique requires LLMs to compare pairs of retrieved tuples and determine which one is more relevant to the incomplete tuple, as shown in Figure 3 (b). Unlike the listwise strategy that reranks an entire list, the pairwise method focuses on one-to-one comparisons, making it a less complex task.

---

[2]https://huggingface.co/bert-base-uncased
[3]https://huggingface.co/castorini/monot5-base-msmarco-10k

## C   PROMPT AND EXAMPLE OF REASONER

For the final reasoning step, leveraging the advanced capabilities of LLMs, we employ a template-driven approach to structure prompts that guide the LLMs in generating the necessary outputs, as shown in Figure 4.

Given the retrieved tuples, LLMs can accurately infer missing values and provide explanations on how the selected tuples aid in imputation if users require insight of the reasoning process. Even in cases where no relevant tuples are retrieved, the powerful LLMs can accurately determine the lack of relevance and refuse to imputation. Moreover, users can quickly assess the accuracy of the imputed values based on the small number of retrieved tuples and the model's explanations.

It is critical to note that simply increasing the number of retrieved tuples does not necessarily enhance the performance of LLMs. The effectiveness of LLMs is significantly influenced by the amount of contextual data provided (Liu et al., 2023). While more retrieved tuples may increase the chance of including relevant tuples, it also burdens the model with excessive information, potentially diminishing reasoning accuracy. In Section 4, we conduct comparative experiments to explore how varying the number of retrieved tuples affects data imputation accuracy. These experiments substantiate our rationale for selecting only the top-5 retrieved tuples and highlight the importance of employing a reranker to optimize outcomes. It also demonstrates the advantage of tuple-level retrieval over table-level one by significantly reducing the number of irrelevant tuples.

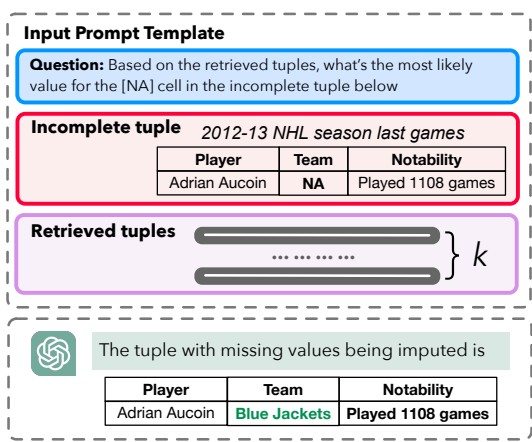

Figure 4: Prompt template for reasoner.

Table 7 presents an example of our input to the GPT-4 and its corresponding output. The input prompt basically follows the template structure described above, while we mandate the output format as JSON for processing outputs in batches and evaluating performance effectively. In this example, both the "district" and "party" values are missing. However, the retrieved tuple 1 contains the party information (D) and district information (OH-19) for Eric Fingerhut. We can see that from the output, the LLM successfully identifies the information needed to fill in the missing values and understands that "D" srepresents Democratic.

## D   EXAMPLES OF SYNTHESIZING TRAINING DATA

Table 8 presents three examples of the synthesized training data used for the retriever model. Each example consists of an anchor tuple, a positive tuple, and two negative tuples. For each tuple, we provide the corresponding caption and attribute/value pairs. In the actual training data, each anchor tuple is paired with seven negative tuples. while for brevity, only two negative tuples are shown here, as they are sufficient to illustrate the characteristics of the negative samples.

## E   DETAILS OF ABLATION STUDY

### E.1   NUMBER OF RETRIEVED TUPLES V.S. IMPUTATION PERFORMANCE

Intuitively, providing more retrieved tuples to LLMs seems beneficial as the chance of retrieving relevant tuples increases with the number of tuples ($k$). However, the expanded input length also introduces complexities in the LLMs' reasoning. To explore the impact of $k$ on data imputation, we conduct experiments using all test tuples in the Show Movie and Cricket Players datasets and 500 sampled test tuples from the other three datasets. We feed various number of tuples retrieved by our reranker into GPT-3.5, analyzing changes in data imputation accuracy and retrieval success rate. To minimize costs, we do not use GPT-4.

Table 7: An example of reasoning step

**Input:** Based on the retrieved tabular data, what's the most likely value for the [NA] cell in the table below? Please respond using JSON: {district(s): "", party: ""}, the key is attribute name of each [NA], value is the predicted value for each [NA].

*Missing Tuple*:

caption: List of living former members of the United States House of Representatives

| representative | state | district(s) | served | party | date of birth | age |
| Eric Fingerhut | Ohio | [NA] | 1993–1995 | [NA] | May 6, 1959 | 54 years, 271 days |

*Retrieved Tuples*:

Tuple 1: caption: list of members of the united states house of representatives in the 103rd congress by seniority

| rank | representative | party | district | seniority date | notes |
| 362 | Eric Fingerhut | D | OH-19 | January 3, 1993 | Left the House in 1995 |

Tuple 2: caption: united states house of representatives elections, 1994 Ohio

| district | incumbent | party | first elected | status | opponent |
| Ohio19 | Eric Fingerhut | Democratic | 1992 | Defeated Republican gain |
Steve LaTourette (R) 48.5% Eric Fingerhut (D) 43.5% Ron Young (I) 5.5% Jerome A. Brentar (I) 2.5% |

Tuple 3: caption: republican revolution house of representatives

| name | district | predecessor | predecessor's fate |
| Steve LaTourette | Ohio-19 | Eric Fingerhut | Defeated |

Tuple 4: caption: list of stanford university people members of the u.s. house of representatives

|name | class year | notability |
| Eric Fingerhut | J.D. 1984 | U.S. Congressman |

Tuple 5: caption: alpha epsilon pi notable alumni

| name | original chapter | notability |
| Eric Fingerhut | Honorary | Ohio State Senator, Chancellor of Ohio State Board of Regents |

**Output:** {"district(s)": "OH-19","party": "Democratic"}

Table 8: Examples of synthesizing training data.

| Example 1 | |
|---|---|
| Anchor Tuple | **caption:** new york film critics circle award for best actress 1990s |
| | (year: 1993, winner: Holly Hunter, film: The Piano, role: Ada McGrath) |
| Positive Tuple | **caption:** york film critics best award for traffic_circle actress 1990 |
| | (role: Ada McGrath, winner: Holly Hunter) |
| Negative Tuple - 1 | **caption:** new york film critic award for actress best 1990s |
| | (winner: Jodie Foster) |
| Negative Tuple - 2 | **caption:** critics take actress award york best new circle 1990s |
| | (year: 1995, role: Sadie Flood, winner: Jennifer Jason Leigh) |
| **Example 2** | |
| Anchor Tuple | **caption:** 1992 texas rangers season farm system |
| | (level: Rookie, team: GCL Rangers, league: Gulf Coast League, manager: [MASK]) |
| Positive Tuple | **caption:** 1992 texas rangers season farm system |
| | (level: Rookie, team: GCL Rangers, league: Gulf Coast, manager: Chino Cadahia) |
| Negative Tuple - 1 | **caption:** TX rangers season farm system |
| | (team: Tulsa Drillers, league: Texas League, manager: Bobby Jones) |
| Negative Tuple - 2 | **caption:** 1983 texas rangers season farm system |
| | (level: AA, team: Tulsa Drillers, league: Texas League, manager: Marty Scott) |
| **Example 3** | |
| Anchor Tuple | **caption:** AFI's 10 top 10 romantic comedy |
| | (#: 6, film: When Harry Met Sally..., year: [MASK]) |
| Positive Tuple | **caption:** academy award for best writing (original screenplay) 1980s |
| | (year: 1989 (62nd), film: When Harry Met Sally..., screenwriter(s): ) |
| Negative Tuple - 1 | **caption:** AFI's superlative 10 romanticist |
| | (year: 1931, film: City Lights) |
| Negative Tuple - 2 | **caption:** AFI's 100 years...100 laughs the list |
| | (#: 22, movie: Adam's Rib, director: George Cukor, year: 1949) |

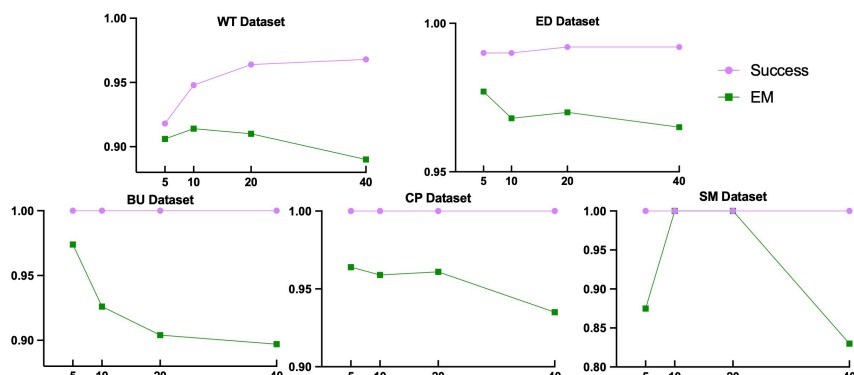

Figure 5: Performance *vs.* #-retrieved tuples. X-axis: number of retrieved tuple sent to LLM. Y-axis: evaluation scores.

Figure 5 reveals that data imputation accuracy does not significantly increase with the rising number of retrieved tuples fed to LLMs. Instead, a decrease in accuracy is evident across all datasets, with a marked decline when $k$ reaches 40, suggesting that an excess of irrelevant tuples can severely affect LLM's imputation accuracy. This outcome highlights a trade-off: an increase in the number of retrieved tuples sent to LLMs may inversely impact data imputation accuracy. This underscores the importance of an efficient retrieval module capable of precisely identifying relevant tuples with the smallest possible $k$ and highlights the advantages of tuple-level over table-level retrieval.

### E.2   NUMBER OF EXAMPLE TUPLES V.S. IMPUTATION PERFORMANCE

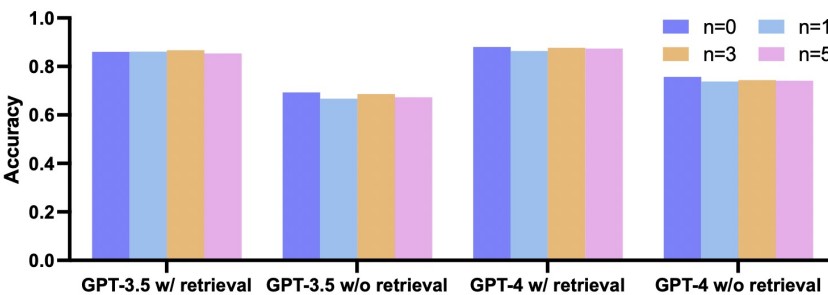

Figure 6: Accuracy *vs.* Number of Examples

The performance of LLMs is closely tied to the context they receive. Besides retrieved tuples, another important factor that may influence performance is the example, *i.e.,* complete tuple, which can guide the model towards the domain and format of the missing value. For instance, in Table 7, for the tuple with missing values (Eric Fingerhut, Ohio, [NA], 1993-1995, . . . ), we can include another complete tuple, such as (Dan Miller, Florida, Florida's 13th congressional district, 1993–2003, . . . ) as an example. Intuitively, the example tuple can prompt the model with information about the domain and format of the missing content.

To investigate the impact of examples, we randomly sample 200 incomplete tuples with missing values from the WikiTuples test set. For each tuple, we select $n$ ($n \in 0, 1, 3, 5$) complete tuples from the same table as examples, keeping the other inputs the same as in Figure 4.

Figure 6 shows that, surprisingly, no clear correlation between the number of example tuples and imputation accuracy. This might be attributed to the fact that the tables in our scenario inherently lack redundant data, making it difficult to infer missing values solely based on the complete tuples within the table.

### E.3   IMPACT OF SYNTHESIZED TRAINING DATA

This experiment is designed to evaluate the impact of different methods of synthesizing training data on the performance of our retriever. We focus on three main aspects: anchor tuples, positive tuples, and negative tuples. The results of this coparative experiment are presented in Table 9.

Table 9: Performance of retriever *vs.* Training datasets.

| Training Datasets | | WT | SM | ED | CP | BU |
|---|---|---|---|---|---|---|
| **RAI** | | **0.945** | **1.0** | **0.992** | **1.0** | **1.0** |
| Anchor | w/ complete tuples | 0.936 | 0.958 | 0.985 | 1.0 | 1.0 |
| | w/ missing values | 0.946 | 1.0 | 0.976 | 0.966 | 1.0 |
| Positives | w/o tuples from other tables | 0.924 | 1.0 | 0.962 | 0.967 | 1.0 |
| Negatives | w/ augmented anchor without masked cells | 0.862 | 0.833 | 0.93 | 1.0 | 1.0 |

**Anchor Tuples.** In synthesizing training data, we employ two types of anchor tuples: complete tuples and tuples with cells masked (*i.e.,* simulating missing values). To show that combining these two types of anchor tuples yields better results, we reconstruct training data considering: anchor tuples consisting of only complete tuples and anchor tuples consisting of only tuples with masked cells. From the row 1 to 3, it is evident that combining complete anchor tuples with those having missing values enhances retriever performance.

**Positive Tuples.** Unlike traditional methods (Wang et al., 2023) that only consider augmented anchors as positives, we include relevant tuples from other tables in our synthesized data. Removing such positives (row 4) leads to a significant decline in the retriever's performance, highlighting the importance of diverse positive samples with various heterogeneous attributes, which aligns with real-world scenarios.

**Negative Tuples.** We only regard other tuples from the anchor's table as hard negatives. However, since our retriever works on data imputation, for an anchor tuple $t$ with $t[j]$ masked, it's intuitive to consider augmented anchors with the $j$-th attribute deleted as additional negatives. These tuples are very similar to the anchor but lack information to assist in imputing missing values. To test the effectiveness of this intuitive approach, we add this type of hard negative to the training data and report the corresponding result in row 5.

Surprisingly, we observe that it leads to a significant decrease in retrieval results. We hypothesize that this is mainly because it is challenging to distinguish cell-level semantics accurately when encoding each tuple into an embedding. This variation causes the most substantial decrease in performance compared to other changes in positives and anchors.

## F   CONSTRUCTION OF MVBENCH

### F.1   DATASETS COLLECTION AND CONSTRUCTION

We target datasets that exhibit specific characteristics: (1) they should be derived from real-world scenarios, and (2) they should cover a diverse range of data domains (*e.g.,* business). Also, we particularly focus on datasets that require missing value imputation involving external sources, a challenge that's not well addressed by existing work that typically rely on the inherent data redundancy within the table itself.

Guided by these criteria and informed by existing work in missing value imputation (Deng et al., 2022; Ahmad et al., 2023), we collect five datasets from real-world data sources. By including these challenging scenarios in our benchmark, we demonstrate the effectiveness and generalization of our tuple-level RAG approach.

**WikiTuples.** Based on WikiTables-TURL (Deng et al., 2022), a large collection of high-quality Wikipedia tables, we construct a data lake and incomplete tuples using the train and test sets of WikiTables-TURL respectively.

**Show Movie and Cricket Players.** These two dataset are sourced from RetClean (Ahmad et al., 2023), which provides original tables and corresponding dirty columns in two domains: Cricket Players and Shows Movies. For each domain, we select one table to create tuples with missing values and store tuples from other tables in the data lake.

**Education and Business.** From two sections of the Chicago Data Portal [4] - Education, and Community & Economic Development, we collect tables about school information and business information in the two sections respectively to construct these two datasets. Similarly to Show Movie and Cricket Players, we select one or two tables to create incomplete tuples with missing values while saving the rest in the data lake.

### F.2 RELEVANT TUPLE ANNOTATION

After collecting original datasets and constructing incomplete tuples with data lakes, we start to annotate the relevant tuples for each incomplete tuple. The process involves two steps: Candidate Tuples Construction and Expert Annotation.

**Candidate Tuples Construction.** For an incomplete tuple, we first create a candidate set by selecting tuples that could potentially help in imputing its missing values. We use explicit information, such as similar cell values or cells linking to the same entity, to establish effective filtering rules.

**Expert Annotation.** For each incomplete tuple, we present it and its candidate one by one to a human expert to judge whether the candidate can fill at least one missing value in the incomplete tuple, *i.e.,* be identified as relevant. Before starting the annotation, experts are instructed to apply their domain knowledge carefully. For example, they are advised that certain attributes like a movie director do not change over time, whereas others, such as a sports team manager, may change and thus the temporal alignment between the candidate tuple and the incomplete tuple should be verified. This manual process is time-consuming and requires specific domain knowledge. We hire 10 PhD students as our "human experts" to annotate candidate tuples corresponding to each incomplete tuple, ensuring the quality of our labels. To reduce the cost of annotation, we primarily focus on cases where the candidate set comprised 10 or fewer tuples. In total, over 200 human hours and approximately $1,000 were spent on curating relevant tuple labels for each incomplete tuple.

Our `mvBench` is distributed under the Apache License 2.0, which permits use, distribution, and reproduction in any medium, provided the original work is properly cited and is not used for commercial purposes.

---

[4] https://data.cityofchicago.org/

