# OpenReview forum: "Retrieval Augmented Imputation using Data Lake Tables"
_ICLR.cc/2025/Conference — ICLR 2025 Conference Withdrawn Submission_

### Official Review · Reviewer_6hNh · 2024-10-27

**Soundness:** 2
**Presentation:** 2
**Contribution:** 2
**Rating:** 5
**Confidence:** 4

**Summary:**

proposes a RAG-based approach that retrieves relevant rows from the data lake and feeds the retrieved rows, together with the row for which imputation is required to an LLM to impute the missing value. The paper proposes a new benchmark dataset for training and evaluation of imputation in this context. It presents experimental results showing the proposed approaches improve retrieval of relevant rows, as well as the accuracy of the final imputation.

**Strengths:**

- Using data lakes for imputation is an interesting problem setting-
- The proposed method shows improvements in retrieval
- A new benchmark dataset is introduced

**Weaknesses:**

Technical novelty and contributions. There are two novel ideas in the paper. (1) using data lakes to perform imputation and (2) a new proposed row-level retrieval method for data imputation. However, neither of the two is particularly new and the paper makes limited contributions overall.
- Regarding (1), the paper’s claims on how it’s positioned compared to related works are somewhat contradictory. It mentions that there is a lack of redundancy in their setting, but redundancy is indeed what the model uses for imputation, at least based on the provided examples. The model looks for the same information stated in other available tables. Besides, the paper needs to motivate the problem setting, i.e., in what real-world application are there other data tables stating the same information? Finally, I don’t see any real difference in the problem setting compared with question answering over tables. That is, why can’t I just turn the row with the missing value into a natural language question (which is in fact what the method does) and use existing table QA methods to find the answer? The paper should better clarify what exactly is new in their problem formulation.
- Regarding (2), the training process uses very similar ideas to Sudowoodo. Specifically, the main contribution of the paper is the synthetic data generation, but the proposed ideas are very similar to the data augmentation procedure in Sudowoodo. The paper should clarify the technical differences between the methods.

Experiments. Experiments don’t provide a thorough evaluation of the method. This includes missing baselines and ablation studies as well as insufficient description of the experimental  procedure. As such, the experiments don’t show if the method is truly beneficial, and if so, what novel contribution led to the benefits.
- Why not include Sudowoodo in Table 4? Given the large discrepancy between the retrieval accuracy of BM25 (table 5) and its final imputation accuracy (table 4), presenting Table 4 with Sudowoodo can provide a better understanding of the method's contributions
- Using pre-trained embedding models (e.g., openai’s embedding model) is a common retrieval method. The paper should compare against embedding rows using pre-trained embedding models.
- Do Sudowoodo and RAI have the same base bert model? Are they both using an existing pre-trained model? Having the same starting point should help better understand differences between the two. The paper should also report the accuracy for the (pre-trained) BERT model before fine-tuning with their method
- The paper needs to provide a better and thorough description of the train/test splits used. Is the retriever training only done on the WT dataset? How different is the WT dataset from other datasets? Given that WT contains Wikipedia tables, can questions in CM and CP be answered based on the information in WT? It is important to understand if there is in fact any domain difference between the datasets
- I don’t understand why the datasets need to be manually labeled, nor why a human is expected to be able to provide correct labels (given that not all the information may be known by a person).   Why not take a complete table, drop some values from some of the cells, and use the dropped values as ground-truth answers?
- The paper should use an entity linking method for retrieval. The paper states that such a method is used during training to find positive samples. Why not do that at test time? Given that the workload seems to be mainly finding information about an existing entity, perhaps entity linking should be used for retrieval (i.e., link entities, and at query time retrieve entities that are linked)

**Questions:**

Please provide answers to the questions raised above specifically:
- Clarify novelty in problem setting and solution
- Perform the requested experiments or discuss why they cannot/should not be done
- Provide the requested details about the experimental setting

---

### Official Review · Reviewer_D1Lo · 2024-11-03

**Soundness:** 2
**Presentation:** 2
**Contribution:** 2
**Rating:** 5
**Confidence:** 4

**Summary:**

The authors propose Retrieval-Augmented Imputation (RAI), a method for filling missing values in tables by retrieving relevant tuples from a data lake, particularly effective in cases with limited data redundancy. The method utilizes a tuple encoder and a tuple reranker to find tuples containing information related to the missing value. Additionally, the authors propose a method to enhance retrieval accuracy by augmenting the training dataset. Specifically, they augment the training data by modifying the caption, attribute, and value of the tuples. Lastly, they present mvBench, a benchmark for retrieval-augmented imputation, mvBench, to facilitate further research. RAI achieves a 10.7% improvement over existing state-of-the-art methods.

**Strengths:**

S1. The authors address an important issue in data science applications: imputing missing values in tables.

S2. The authors show that their proposed method achieves a 10.7% improvement over existing state-of-the-art methods for table-based retrieval-augmented imputation.

S3. The authors create a new benchmark for data imputation called mvBench.

**Weaknesses:**

W1. The rationale for the Tuple-Level Retrieval is unclear.
The authors' claim that missing values can be imputed with just a few relevant tuples is not convincing, as it contradicts existing methodologies that emphasize the need for a large number of tuples to accurately identify substitute values. The authors should provide references or analysis to support their claims.

W2. The explanation of the process for synthesizing the training dataset is insufficient.
The authors do not specify how each augmentation operator is chosen and applied during the data synthesis. For example, concerning the replace operator, the authors should clarify the criteria for selecting words to replace and how synonyms are identified.

W3. The experimental evaluation is limited.
- The authors should provide end-to-end data imputation accuracy using other retrievers such as Contriever, DPR-scale, BERT with MLM task, and Sudowoodo. This is necessary to demonstrate that RAI's retrieval improves data imputation performance compared to these methods.
- A more detailed explanation is needed to clarify why testing RATA on mvBench is difficult. The authors claim that applying RATA to mvBench is challenging due to varying definitions of "relevant tables." However, this explanation lacks sufficient clarity.
- The authors should conduct an ablation study on the synthesized dataset.
- The authors should explore the sensitivity of different values of K on the retriever's performance to provide a more comprehensive evaluation.

**Questions:**

Please refer to W1, W2, and W3.

---

### Official Review · Reviewer_gySx · 2024-11-04

**Soundness:** 3
**Presentation:** 4
**Contribution:** 2
**Rating:** 3
**Confidence:** 4

**Summary:**

This paper studies data imputation using LLMs. It proposes an RAG-based solution called RAI, consisting of a tuple encoder and a tuple reranker. A benchmark named mvBench is also proposed. The experiments on the benchmark demonstrate the effectiveness and the superiority over state-of-the-art imputation methods.

**Strengths:**

S1. The construction of the training dataset is interesting.

S2. The paper features a benchmark, which can be useful for future research.

S3. The experiments are extensive, with promising experimental results presented.

**Weaknesses:**

W1. The targeted problem (imputation only) is less significant compared to the tasks LLMs are often used for. LLMs (e.g., Table-GPT) can handle various table tasks. It is unclear how the proposed techniques generalize to those other than imputation.

W2. The design of the proposed techniques is routine. Table retrieval for RAG has been explored in previous works, as state in the submission, rendering the proposed framework less novel. The main contribution resides in its tuple encoding and the construction of the training dataset, while the reranking is rather straightforward.

W3. While there are words like "efficient" and "efficiently" in the introduction, I did not find any efficiency evaluation in this paper. Lack of such evaluation might compromise the proposed method's practical use in real-world applications because data lakes are usually heterogenous, noisy, and very large.

W4. Only GPT models are evaluated. It is unknown how the proposed method benefits other models, especially open models, which I believe are more useful for handling large-scale business data due to privacy concerns.

**Questions:**

Q1. I wonder how Challenge 3 (enhancing reasoning with domain knowledge) is addressed in the paper. It seems that the LLM's reasoning ability is simply used here.

Q2. Have you observed the usefulness of your RAG method in other tasks? I think the techniques are quite general and can be applied to other table tasks as well, but they are not explored in the submission.

**Details Of Ethics Concerns:**

This paper studies an RAG approach. I didn't find anything that needs an ethics review.

---

### Note · Authors · 2024-12-03

I have read and agree with the venue's withdrawal policy on behalf of myself and my co-authors.